# A Diet High in Processed Foods, Total Carbohydrates and Added Sugars, and Low in Vegetables and Protein Is Characteristic of Youth with Avoidant/Restrictive Food Intake Disorder

**DOI:** 10.3390/nu11092013

**Published:** 2019-08-27

**Authors:** Stephanie G. Harshman, Olivia Wons, Madeline S. Rogers, Alyssa M. Izquierdo, Tara M. Holmes, Reitumetse L. Pulumo, Elisa Asanza, Kamryn T. Eddy, Madhusmita Misra, Nadia Micali, Elizabeth A. Lawson, Jennifer J. Thomas

**Affiliations:** 1Neuroendocrine Unit, Massachusetts General Hospital, Boston, MA 02114, USA; 2Eating Disorders Clinical and Research Program, Massachusetts General Hospital, Boston, MA 02114, USA; 3Translational and Clinical Research Center, Massachusetts General Hospital, Boston, MA 02114, USA; 4Department of Psychiatry, Harvard Medical School, Boston, MA 02114, USA; 5Department of Medicine, Harvard Medical School, Boston, MA 02114, USA; 6Département universitaire de psychiatrie, Université de Genève, 1211 Genève, Switzerland; 7Service de psychiatrie de l’enfant et de l’adolescent, Département de l’enfant et de l’adolescent, HUG, 1211 Genève, Switzerland; 8Department of Pediatrics, Gynecology and Obstetrics, University of Geneva, 1211 Geneva, Switzerland; 9Great Ormond Street Institute of Child Health, University College London, 30 Guilford Street, Holborn, London WC1N 1EH, UK

**Keywords:** avoidant/restrictive food intake disorder, ARFID, dietary recall, dietary intake, nutritional insufficiency

## Abstract

Avoidant/restrictive food intake disorder (ARFID) is characterized in part by limited dietary variety, but dietary characteristics of this disorder have not yet been systematically studied. Our objective was to examine dietary intake defined by diet variety, macronutrient intake, and micronutrient intake in children and adolescents with full or subthreshold ARFID in comparison to healthy controls. We collected and analyzed four-day food record data for 52 participants with full or subthreshold ARFID, and 52 healthy controls, aged 9–22 years. We examined frequency of commonly reported foods by logistic regression and intake by food groups, macronutrients, and micronutrients between groups with repeated-measures ANOVA. Participants with full or subthreshold ARFID did not report any fruit or vegetable category in their top five most commonly reported food categories, whereas these food groups occupied three of the top five groups for healthy controls. Vegetable and protein intake were significantly lower in full or subthreshold ARFID compared to healthy controls. Intakes of added sugars and total carbohydrates were significantly higher in full or subthreshold ARFID compared to healthy controls. Individuals with full or subthreshold ARFID had lower intake of vitamins K and B12, consistent with limited vegetable and protein intake compared to healthy controls. Our results support the need for diet diversification as part of therapeutic interventions for ARFID to reduce risk for nutrient insufficiencies and related complications.

## 1. Introduction

Avoidant/restrictive food intake disorder (ARFID) was recently added to the Diagnostic and Statistical Manual of Mental Disorders, 5th edition (*DSM-5*) Feeding and Eating Disorders section to replace and expand upon the former *DSM-IV* diagnosis of feeding disorder of infancy and early childhood. Individuals with ARFID have a unique psychological profile manifesting as sensory sensitivity, fear of aversive consequences (e.g., choking or vomiting), and/or lack of interest in food or eating. Yet, they often present with dysfunctional feeding behaviors similar to anorexia nervosa (AN), including restrictive eating [1]. Restrictive eating in ARFID is typically characterized not only by limited volume, but also limited variety. Case reports suggest that individuals with ARFID often limit their intake to highly palatable carbohydrate-rich processed foods, and—in more extreme cases—only eat foods within two food groups (e.g., grains and dairy), or even just one or two foods total (e.g., pasta and bread) [2]. This severely selective diet can lead to nutrition-related medical issues such as electrolyte imbalances, fat-soluble vitamin deficiencies, B vitamin deficiencies (including vitamin B12 and folate deficiencies), and mercury toxicity [2,3,4]. Unique to this population is the prevalence of nutritional deficiencies that exist independent of low weight or even in the absence of faltering growth [5,6,7].

It is well known that a nutritionally adequate and balanced diet is critical for optimal growth and development in children and adolescents. The U.S. Department of Agriculture (USDA) Dietary Guidelines for Americans promote the intake of a diverse diet including nutrient-dense foods such as varied fruits and vegetables, whole grains, and lean proteins. Specific nutrient reference values for over 40 nutrients are described in the Dietary Reference Intakes (DRI) developed by the Institute of Medicine of the National Academies. The DRI are intended to guide nutrition recommendations and target adequate intake of both macronutrients and micronutrients [8,9,10]. These recommendations further support the need for a varied diet as individual foods even within a food group can differ with respect to nutritional composition [11]. Current estimates of nutrient status in the U.S. population indicate that vitamins A, C, D, E, folate, B6, B12, calcium, zinc, iron, and magnesium are often under-consumed relative to their DRI [12,13]. Coupling this general estimate of under-consumption with disordered eating may increase risk of deficiencies in the ARFID population.

The USDA Dietary Guidelines for Americans recommend that both children and adults limit consumption of saturated fat, added sugar, and added salt, which are most often found in processed foods that may displace healthier alternatives. Processed foods—defined as any food that is altered during preparation (i.e., baking, drying, freezing)—include cereals, bread, snacks (i.e., chips, pies, pastries), convenience foods such as ready-to-eat meals, and sugar-sweetened beverages [14,15]. Globally (and in the U.S.), processed foods are the predominant source of nutrients and energy accounting for 50%–90% (and more than 70%, respectively) of intake [16,17]. Greater intake of processed foods is associated with health consequences including obesity, type 2 diabetes, and cardiovascular disease [18,19], supporting the importance of limiting processed foods and promoting diet quality with nutrition education and lifestyle interventions.

Clinical observations suggest that individuals with ARFID often rely heavily on processed foods and limit their intake of fruits and vegetables, which may contribute to failure to meet adequate nutritional and/or energy needs with associated negative health implications. Few studies have evaluated the role of picky eating, a unique restrictive eating pattern that can lead to symptoms of ARFID, in children or adults [20,21]. Among individuals with picky eating, severity of picky eating has been inversely correlated with diet variety, more specifically, intake of fruits and vegetables [22,23,24]. However, dietary intake has yet to be systematically examined in the ARFID population. To empirically characterize dietary intake patterns associated with ARFID, we evaluated differences in four-day food record data in children and adolescents with full or subthreshold ARFID compared to healthy controls. This is the first study to assess dietary patterns, food intake by food groups, and usual intake of energy, macronutrients, and micronutrients of individuals with full or subthreshold ARFID. We hypothesized that, consistent with our clinical observations, individuals with full or subthreshold ARFID would commonly report intake of foods that are defined as processed foods, leading to greater consumption of added sugars and fat. We also hypothesized that greater consumption of processed foods would displace healthier food groups, such that individuals with full or subthreshold ARFID would report lower intake of fruits, vegetables, and fibrous grains compared to healthy controls. We further predicted that these differences in intake among food groups would be reflected in differential macronutrient intake such that individuals with full or subthreshold ARFID would have a diet higher in carbohydrates and fat (but lower in protein) compared to healthy controls. Lastly, we hypothesized that individuals with full or subthreshold ARFID would have significantly lower intake of fat-soluble (vitamins A, D, E, and K) and water-soluble vitamins (vitamins B6, folate, B12, and C) and minerals (calcium, iron, magnesium, zinc) compared to healthy controls, with a greater percentage of full or subthreshold ARFID not meeting the current dietary recommendations for micronutrient intake.

## 2. Materials and Methods

We collected dietary recall data as part of ongoing observational studies examining the neurobiological mechanisms of feeding and eating disorders in youth (NIH R01 MH108595, R01 MH103402, and Harvard Catalyst grant M01-RR-01066). We analyzed data collected between 2008 and 2018.

### 2.1. Participants

Our participants with full and subthreshold ARFID included males and females aged 9–22 years (*n* = 52) who met criteria for ARFID on the Eating Disorder Assessment-5 (EDA-5) [25] or endorsed significant ARFID symptoms on the Kiddie Schedule for Affective Disorders and Schizophrenia for School-Aged Children–Present and Lifetime version (KSADS-PL) Eating Disorder and Substance-Related Disorders Supplements [26]. We used the four *DSM-5* criteria and the Pica, ARFID, and Rumination Disorder Interview (PARDI) to define full or subthreshold ARFID [27,28]. More specifically, we defined full ARFID as individuals who restricted their intake by volume and/or variety and met *DSM-5* criteria A (i.e., weight loss, nutritional deficiency, dependence on enteral feeding or oral nutrition supplements, and/or marked interference with psychosocial function), B (the eating disturbance is not explained by lack of available food or a culturally sanctioned practice), C (the eating disturbance does not occur exclusively during the course of AN or bulimia nervosa, and there is no evidence of shape and weight concerns), and D (the eating disturbance is not attributed to a concurrent medical condition or psychiatric disorder) for ARFID, as suggested in the PARDI diagnostic algorithm. Those with subthreshold ARFID restricted their intake by volume and/or variety and met *DSM-5* criteria B, C, and D, but did not meet criteria A1–A4 at the level of severity required by the PARDI. For the full and subthreshold ARFID group, exclusion criteria included any co-occurring psychotic disorders; gastrointestinal tract surgery; medical history of intellectual disability (IQ < 70); history of psychosis; active substance or alcohol use disorder within the past month; and active suicidal ideation.

We drew healthy control participants (*n* = 52) from three studies with slightly different enrollment criteria. More specifically, study inclusion criteria for healthy controls included males and females aged 9–22 years, who completed a health history review and physical examination, and screening laboratory tests to confirm sound physical health, a body mass index (BMI) within the 25th–85th percentile for age, and regular menses after the first two years post-menarche if female. Exclusion criteria for the healthy control participants included psychological disorders determined by one of the following: Children’s Depression Rating Scale-Revised (CDRS-R), Children’s Depression Inventory (CDI) [29,30,31,32], or KSADS-PL [26]; feeding or eating disorders determined by one of the following: EDA-5 [25], Eating Disorder Examination Questionnaire (EDE-Q) [33], or extensive review of health history and physical examination; gastrointestinal tract surgery; any neuroendocrine abnormalities or conditions; active substance or alcohol use disorder within the past month; and active suicidal ideation.

All psychiatric interviews were administered by doctoral-level psychologists or trained research coordinators, who participated in weekly inter-rater reliability meetings chaired by a psychologist. The Partners Human Research Committee approved data collection for all studies from which participants were drawn. We obtained informed consent from adult participants (ages 18 and older) and parent/guardian consent plus child assent for child participants (ages 9–17).

### 2.2. Anthropometric Measurements

Study staff measured height with a single stadiometer, and weight by an electronic scale in triplicate, then averaged to obtain final values. We calculated BMI as the ratio of weight (kg) to height (m) squared [34]. We calculated percent of median BMI by identifying the median BMI for the following age groups 9–13 years, 14–18 years, and 18+ years of age.

### 2.3. Assessment of Food and Nutrient Intake

We evaluated all participants at either the Translational and Clinical Research Center (TCRC) at Massachusetts General Hospital or the Athinoula A. Martinos Center for Biomedical Imaging. The visit included a review of health history and physical examination, and review of a four-day food record [35,36] that participants completed at home prior to the study visit. Instructions for the four-day food record were provided and reviewed by a research dietitian during screening. Research dietitians entered and analyzed the data utilizing the 2018 version of the Nutrition Data System for Research (NDS-R) software from the Nutrition Coordinating Center (NCC) at the University of Minnesota [37,38,39]. A research dietitian reviewed all data entered for analysis and final reports. Vitamin and mineral supplements (e.g., over-the-counter multivitamin and multimineral supplements, individual vitamin and/or mineral supplements, and nonprescription antacids) were not incorporated into the food and nutrient intakes used for statistical analysis, as our objective focused on evaluating intake from food specifically, rather than supplements. Use of calorie-containing nutrient/energy supplements including Ensure^®^, Ensure^®^ Plus, Pediasure^®^, Boost^®^, and Carnation Breakfast Essentials^®^, was captured during the interview with the TCRC registered dietitian at Massachusetts General Hospital or the Athinoula A. Martinos Center for Biomedical Imaging. Additionally, the calorie-containing nutrient/energy supplements were captured as food items according to the NDS-R NCC and analyzed as food items reported in the four-day food record. We used reference values from the USDA Dietary Guidelines for Americans and DRI [12] to determine the percent of participants not meeting a recommendation for food groups and micronutrients including vitamins A, C, D, E, K, folate, B6, B12, calcium, iron, magnesium, and zinc [8,9,10]. To better understand whether our participants exhibited the limited dietary variety previously observed in anecdotal evidence and case reports, we completed a qualitative review of four-day food record data, identifying the five most commonly reported food subgroups defined by the NDS-R NCC Food Group Serving Count System, which categorizes food into 169 subgroups across nine general categories [37,38,39]. We defined frequency of commonly reported foods by the most common subgroups reported at least once in the food record and identified the top five most common subgroups for the full or subthreshold ARFID group as well as the top five most common subgroups for the healthy control group. We entered data as ‘present/absent’ in the food record for the five subgroups identified within the full or subthreshold ARFID and healthy control groups.

### 2.4. Statistical Analyses

To determine differences among clinical characteristics between the participant classification groups, we used Analysis of Variance (ANOVA) for continuous variables including age, percent of median BMI, and multinomial or logistic regression for categorical variables including sex, ethnicity, and race. Results of this analysis identified potential confounding variables including age and sex. Thus, we adjusted all subsequent analyses presented for age and sex.

We compared differences in the frequency of the most commonly reported food subgroups defined by the NDS-R NCC Food Group Serving Count System between full or subthreshold ARFID and healthy controls by logistic regression.

To determine the usual distributions among (a) intake of food groups (measured in cup equivalents, ounces, or teaspoons), (b) energy and macronutrient intake, and (c) micronutrient intake, we utilized a measurement-error model based on a linear regression adjusting for intra-person day-to-day variability, followed by a Box–Cox transformation to skewed data distributions [40]. Intake by food groups, macronutrient intake, and micronutrient intake were analyzed by a repeated-measures ANOVA for non-skewed distributions, or the nonparametric Wilcoxon’s Rank Sum test for skewed distributions, adjusting for age and sex. All models were assessed using diagnostics for assumptions of homogeneity of variance. We found no outliers that influenced significance in the full models. We used the Bonferroni–Holm procedure to adjust for multiple comparisons, by adjusting the rejection criteria for each hypothesis [41,42]. Level of significance for each hypothesis was as follows: individuals with full or subthreshold ARFID would commonly report intake of foods that were defined as processed foods, *p* < 0.01; individuals with full or subthreshold ARFID would report lower intake of fruits, vegetables, and fibrous grains compared to healthy controls, *p* < 0.01; individuals with full or subthreshold ARFID would have a diet higher in carbohydrates and fat (but lower in protein) compared to healthy controls, *p* < 0.0125; individuals with full or subthreshold ARFID would have significantly lower intake of fat-soluble vitamins (vitamins A, D, E, and K), water-soluble vitamins (vitamins B6, folate, B12, and C), and minerals (calcium, iron, magnesium, zinc) compared to healthy controls, *p* < 0.0125. We carried out all analyses using SAS v 9.4 (Cary, NC). Data are reported as means ± standard error of mean (SEM).

## 3. Results

### 3.1. Participant Characteristics

Participant characteristics are summarized in Table 1. The mean ages of participants with full or subthreshold ARFID and healthy controls were 14.3 ± 0.4 years and 16.9 ± 0.4 years, respectively. Healthy controls were approximately 2.6 years older than participants with full or subthreshold ARFID. Males comprised 62% (*n* = 32) of the full or subthreshold ARFID group, compared to 39% (*n* = 20) of healthy controls. Anthropometric measurements were captured as percent of median BMI, which was not significantly different between groups. Ethnicity did not significantly differ between groups. Racial diversity was captured in the following categories: Caucasian, African American, Asian, more than one race, and not specified. Race did not significantly differ between the full or subthreshold ARFID group and healthy controls.

### 3.2. Frequency of Commonly Reported Foods

The most commonly reported foods based on the NDS-R NCC classification among participants with full or subthreshold ARFID and healthy controls are reported in Table 2 and Table 3. In both groups, the most commonly reported foods fell in the category of ‘sugar, syrup, honey, jam, jelly, and preserves’ and ‘cheese, full fat’; however, they did not significantly differ between the two groups (*p* = 0.17 and *p* = 0.10, respectively). Consistent with our expectations, participants with full or subthreshold ARFID reported a significantly higher frequency of intake of the following food categories compared to healthy controls: ‘loaf type bread and plain rolls-refined grains’: 52% vs. 40%; and ‘cakes, cookies, pies, pastries, Danishes, doughnuts, and cobblers’: 48% vs. 35% (all *p* < 0.01). In contrast, healthy controls reported a significantly higher frequency of intake of other vegetables, fruits (excluding citrus fruits), and dark-green vegetables (56% vs. 21%, 46% vs. 25%, and 46% vs. 17% respectively; all *p* < 0.01).

### 3.3. Dietary Intake Among Food Groups

Mean intake among food groups as defined by the USDA Dietary Guidelines for Americans in full or subthreshold ARFID compared to healthy controls is reported in Table 4 [8,9,10]. Dietary intake of vegetables including all fresh, frozen, canned, and dried options in cooked or raw forms, and vegetable juices was significantly lower in full or subthreshold ARFID compared to healthy controls. Approximately 86% of the participants with full or subthreshold ARFID and 65% of healthy controls did not meet the USDA Dietary Guidelines for Americans recommendation of vegetables per day. Additionally, protein intake (including foods from both animal and plant sources) was significantly lower among participants with full or subthreshold ARFID compared to healthy controls, with 76% of participants with full or subthreshold ARFID not meeting the current recommendation for protein intake compared to approximately 50% of healthy controls. Although we found no significant difference in fruit intake between groups, the percent of participants not meeting the recommendation for daily fruit intake was significantly greater in participants with full or subthreshold ARFID compared to healthy controls (72% vs. 57%, *p* = 0.001). We found no significant differences in mean intake in the grains, dairy, and oils food groups, and no significant differences in the percent of participants not meeting recommendations in the grains, dairy, and oils food groups.

### 3.4. Energy and Macronutrient Intake

Absolute calorie intake and total grams consumed reported by participants with full or subthreshold ARFID were comparable to healthy controls (Table 5). Intake of added sugars was significantly higher in participants with full or subthreshold ARFID, with a parallel significant difference in the reported absolute intake of carbohydrates. We observed higher intake of total sugars, and lower intake of fiber, in participants with full or subthreshold ARFID compared to healthy controls; however, the differences were not significant after controlling for multiple comparisons (all *p* > 0.02). Total protein intake was significantly lower in the full or subthreshold ARFID group compared to healthy controls. Additionally, participants with full or subthreshold ARFID consumed a lower percentage of total calories from protein (ARFID: 12.0 ± 0.42; healthy controls: 16.4 ± 0.46, *p* < 0.001) and a greater percentage of calories from carbohydrates (ARFID: 54.3 ± 0.64; healthy controls: 50.7 ± 1.05, *p* = 0.004) compared to healthy controls. Percent of calories from fats (ARFID: 33.5 ± 0.56; healthy controls: 32.4 ± 0.71) did not differ between groups (*p* = 0.185). Among participants with full or subthreshold ARFID, 19% endorsed nutrient/energy supplement use during the interview with a research dietitian, while only six participants (12%) reported foods categorized as ‘nondairy based sweetened meal replacement/supplement’ within the reported four-day food record period. No participants in the healthy control group endorsed nutrient/energy supplement use or reported intake within the four-day food record.

### 3.5. Micronutrient Intake

Dietary intake of vitamin K (as micrograms of phylloquinone) was significantly lower in participants with full or subthreshold ARFID compared to healthy controls (Table 6) [43], with 78% of full or subthreshold ARFID participants (vs. 55% of healthy controls) not meeting the age-appropriate DRI for males and females [8,9,10]. Among the other fat-soluble vitamins, there were no significant differences between groups. Vitamin B12 was the only water-soluble vitamin that significantly differed between participants with full or subthreshold ARFID and healthy controls with lower intake of vitamin B12 in participants with full or subthreshold ARFID. There was no significant difference in the percent of participants not meeting the DRI [8,9,10]. Intake of vitamin B6, vitamin C, folate, calcium, magnesium, and zinc did not differ between groups. However, the percent of participants not meeting the DRI for magnesium and zinc was significantly higher in participants with full or subthreshold ARFID compared to healthy controls (90% vs. 76%; 65% vs. 52%, respectively). Among participants with full or subthreshold ARFID, 41% endorsed multivitamin/multimineral supplement use, compared to 14% of healthy controls during the interview with a research dietitian.

## 4. Discussion

This is the first report systematically characterizing dietary intake in youth with full or subthreshold ARFID. We found that, compared to heathy children and adolescents, the diet of youth with full or subthreshold ARFID is higher in refined-carbohydrate processed foods, as well as total carbohydrates and added sugars, and lower in protein, vegetables, and vitamins K and B12. Our findings demonstrate that overall diet quality of youth with full or subthreshold ARFID is inadequate in macro- and micronutrient composition, ultimately increasing the risk for nutritional deficiencies and metabolic disorders.

We demonstrated that participants with full or subthreshold ARFID have a diet high in added sugars, a major component of processed foods, more specifically, foods categorized as refined grains. Refined grains have been milled, which is a process that removes the bran and germ. This results in a finer texture and improves shelf life, but it also removes dietary fiber, iron, and many B vitamins. Frequent consumption of processed foods can increase risk for metabolic sequelae, as they tend to be higher in added sugars, sodium, and solid fats, and lower in fiber and protein. The high palatability and unique macro- and micronutrient composition of processed foods is thought to contribute to gut-brain signaling dysfunction, which may drive intake and reinforce pathological eating behaviors [44,45,46,47]. Results from a randomized controlled trial found that individuals randomized to a processed food diet had a greater intake of total calories, fat, and carbohydrates, a lower intake of protein, and a significant increase in weight over a two-week period compared to individuals randomized to an unprocessed diet. These findings further support the need to limit consumption of processed foods despite abundant access and ease of purchase and preparation [48]. In children and adolescents, increased intake of processed foods is associated with overweight and obesity, elevated blood glucose levels, and increased risk of hypertension [49,50]. Within our sample, percent of median BMI was not significantly different between groups. However, examining the long-term effects of limited diet variety and intake of processed foods in ARFID is warranted to determine the risk of increasing BMI and associated metabolic dysfunction. Increased consumption of processed foods is related to reduced intake of other food groups including fruits, vegetables, whole grains, and protein [51,52]. We found that most of the commonly eaten foods reported by participants with full or subthreshold ARFID include processed foods and refined carbohydrates and that individuals with full or subthreshold ARFID report these foods at a significantly greater frequency than healthy controls. This likely contributes to the greater intake of total carbohydrates, percent of total calories from carbohydrates and added sugars, and reduced intake of protein and percent of total calories from protein that we observed in individuals with full or subthreshold ARFID.

Similar to AN, some individuals with ARFID are low-weight, which is often associated with inadequate intake as defined by limited volume and calories. However, other individuals with ARFID are of normal weight or overweight and report inadequate dietary variety. Other than in low-weight individuals where energy repletion is a critical first step, our findings suggest that therapeutic interventions for ARFID should prioritize the introduction of nutritionally adequate foods over processed foods with known detrimental effects. While patients with very severe presentations of ARFID may require gradual exposure to novel foods, early changes to eating flexibility that favor minor variations in processed foods or the introduction of supplements should ultimately be leveraged to initiate bigger changes such as the introduction of non-processed whole foods later in treatment [2]. Cognitive-behavioral therapy is one current treatment modality for ARFID, which implements the USDA MyPlate guidelines as well as describes common micronutrient deficiencies to help identify foods for exposure therapy and learning [2]. In addition, multidisciplinary teams including those in gastroenterology, nutrition, speech language pathology, and occupational therapy adopt similar approaches using the MyPlate guidelines as a primary tool for nutrition education and diet expansion [53]. Our findings support current cognitive-behavioral therapy techniques and will help guide future multidisciplinary approaches and development of evidence-based guidelines for treatment specific to ARFID.

Micronutrient analysis revealed that youth with full or subthreshold ARFID reported lower vitamin K (as phylloquinone), reflective of reduced vegetable intake, and lower vitamin B12, consistent with reduced consumption of animal-based proteins, compared to healthy controls [43,54]. Vitamin K is most commonly known for its role in blood clotting; however, novel functions continue to be identified [55]. Vitamin B12 is an essential nutrient for brain development throughout childhood and adolescence due to its involvement in DNA synthesis and methylation, and maintaining genomic stability [54,56]. Vitamin B12 insufficiency is associated with cognitive impairments in children [56]. These findings are relevant for individuals with ARFID—whom we found to have insufficient intake of vitamin K and vitamin B12—and further support the need to maintain a varied diet and meet nutrient requirements from food before supplements. We did not find any other micronutrient insufficiency in this group, which may be attributed to consumption of processed foods. Food processing practices, such as enrichment and fortification, improve the nutrient content in foods by incorporating a variety of fat- and water-soluble vitamins and minerals that help individuals meet the DRI [57,58,59]. With respect to our findings, B vitamins, including vitamin B12, are often found in fortified cereals, whereas vitamin K is not used in enrichment or fortification due to its interaction with anticoagulants [43,54]. Thus, vegetables and plant-based oils remain the primary source of dietary vitamin K [58]. In contrast to previous case reports of micronutrient toxicity in individuals with ARFID due to selective eating [2,3], we found no indication of excessive micronutrient intake. The USDA Dietary Guidelines for Americans and DRI recommend consumption of nutritionally dense foods via diverse dietary intake to ensure optimal nutrient intake for physiological processes and to reduce disease risk [13]. Diet diversity is considered imperative as nutrient composition varies among foods, and the likely synergistic effects among the nutrients and compounds within a food matrix may be lost when relying on supplements [60]. Dependence on nutrient/energy supplements is one way to meet diagnostic criteria for ARFID. While 19% of participants with full or subthreshold ARFID endorsed taking nutrient/energy supplements in general, only six (12%) participants reported actual nutrient/energy supplement use within the four-day food record period, with healthy controls not reporting any use of nutrient/energy supplements. This indicates that supplement use may be sporadic and suggests that providers should maintain vigilance for patient adherence to nutrition-related recommendations. Thus, improving diet diversity in individuals with full or subthreshold ARFID is necessary to ensure nutritional adequacy and to preserve physiological functions independent of a diagnosed nutrient deficiency.

Our findings must be interpreted considering some limitations. First, we chose to use a four-day food record in this first analysis of dietary intake in ARFID; like any self-report instrument, such records are subject to potential bias and misreporting [36]. Second, extensive analysis of micronutrient intake continues to be a challenge as the food composition databases utilized for nutrient composition can be limited in high-quality data representative of current national food habits and consumption patterns [61]. Thus, we limited our micronutrient analysis to vitamins and minerals previously evaluated in pediatric populations. Third, dietary intake of macronutrients and micronutrients is not necessarily a reflection of nutritional status, of which we currently lack well-validated biomarkers [62,63,64]. Thus, dietary questionnaires are useful in identifying risk for nutritional insufficiency. The question of how to best evaluate nutritional status using assessments of food intake or biomarkers continues to be a challenge among clinicians and healthcare practitioners working with ARFID [65]. Our study supports the use of food records in full or subthreshold ARFID as a tool to help guide food exploration and diet diversity within therapeutic interventions.

In summary, in this first study of nutritional intake in youth with full or subthreshold ARFID, we found the diet of youth with full or subthreshold ARFID is higher in refined-carbohydrate processed foods, as well as total carbohydrates and added sugars, and lower in protein, vegetables, and vitamins K and B12. Thus, the severe selectivity and restrictions around eating among individuals with full or subthreshold ARFID appear to increase their risk of nutritional deficiencies. However, a diet high in processed foods may mitigate some micronutrient deficiencies since food processing techniques (e.g., fortification and enrichment) alter the micronutrient content of food. Conversely, a diet high in processed foods increases risk for metabolic diseases. Independent of nutrient status or intake, the selective diet often observed in ARFID may also contribute to psychosocial impairments such as limiting an individual’s ability to engage in social eating with peers, further demonstrating the burden of this disorder. Future studies will be important to examine differences in nutrient intake in individuals with prototypical ARFID presentations (i.e., sensory sensitivity, lack of interest, and fear of aversive consequences) [2]. Our examination of dietary intake in individuals with full or subthreshold ARFID supports the development of therapeutic interventions—such as those utilizing food exposure methodology or cognitive-behavioral therapy—to not only improve overall intake but reduce risk of nutritional deficiencies and lessen the psychosocial burden of the disorder.

## Figures and Tables

**Table 1 nutrients-11-02013-t001:** Clinical characteristics of participants with full or subthreshold avoidant/restrictive food intake disorder (ARFID) and healthy controls.

	Full or Subthreshold ARFID	Healthy Controls	
	*n* = 52	*n* = 52	*p*-Value
Age (y) ^a^	14.3 ± 0.4	16.9 ± 0.4	<0.01 *
Sex (% male)	61.5	38.5	0.05 *
Percent of median BMI (%) ^a^	103 ± 27.8	105 ± 16.6	0.54
Ethnicity (%(*n*))			0.35
Hispanic	10 (5)	22 (11)	
Non-Hispanic	90 (47)	78 (41)	
Racial Diversity (%(*n*))			0.85
Caucasian	92 (48)	62 (32)	
African American	2 (1)	10 (5)	
Asian	2 (1)	11 (6)	
More than one race	4 (2)	13 (7)	
Did not specify	-	4 (2)	

* Significance at *p* < 0.05. ^a^ Data presented as mean ± SEM.

**Table 2 nutrients-11-02013-t002:** Frequency of top five most commonly reported foods among participants with full or subthreshold ARFID compared to healthy controls.

Reported Foods ^a^	Full or Subthreshold ARFID	Healthy Controls	
	(% (*n*))	(% (*n*))	*p*-Value
Sugar, syrup, honey, jam, jelly, preserves	71 (37)	63 (33)	0.17
Cheese, full fat	56 (29)	50 (26)	0.10
Sweetened soft drinks and fruit juice ^b^	54 (28)	46 (24)	0.16
Loaf type bread and plain rolls-refined grains	52 (27)	40 (21)	0.01 *
Cakes, cookies, pies, pastries, Danishes, doughnuts, and cobblers	48 (25)	35 (15)	0.01 *

* Significance at *p*-value < 0.01. ^a^ Foods are summarized into categories defined by the NDS-R NCC Food Group Serving Count System, which categorizes food into 169 subgroups. Frequency of commonly reported foods is defined by the most common subgroups reported at least once in a record. ^b^ Does not include artificially sweetened soft drinks.

**Table 3 nutrients-11-02013-t003:** Frequency of top five most commonly reported foods among healthy controls compared to full or subthreshold ARFID.

Reported Foods ^a^	Healthy Controls	Full or Subthreshold ARFID	
	(% (*n*))	(% (*n*))	*p*-Value
Sugar, syrup, honey, jam, jelly, preserves	63 (33)	71 (37)	0.17
Other vegetables ^b^	56 (29)	21 (11)	<0.01 *
Cheese, full fat	50 (26)	56 (29)	0.10
Fruit (excluding citrus fruit)	46 (24)	25 (13)	<0.01 *
Dark-green vegetables	46 (24)	17 (8)	<0.01 *

* Significance at *p*-value < 0.01. ^a^ Foods are summarized into categories defined by the NDSR NCC Food Group Serving Count System, which categorizes food into 169 subgroups. Frequency of commonly reported foods is defined by the most common subgroups reported at least once in a record. ^b^ Other vegetables (e.g., radishes, vegetable relishes like salsa, and mixed vegetable dishes) not including dark-green vegetables, deep-yellow vegetables, tomato, white potatoes, fried potatoes, other starchy vegetables, legumes, fried vegetables, or vegetable juice.

**Table 4 nutrients-11-02013-t004:** Intake among food groups in participants with full or subthreshold ARFID and healthy controls.

	Mean Intake ^a^			% (*n*) Not Meeting Dietary Guidelines for Americans Recommendation ^b^		
	Full or Subthreshold ARFID	Healthy Controls		Full or Subthreshold ARFID	Healthy Controls	
	*n* = 52	*n* = 52	*p*-Value	*n* = 52	*n* = 52	*p*-Value
Fruit ^c^ (cup equivalents)	1.3 ± 0.1	1.9 ± 0.1	0.03	72 (37)	57 (29)	0.001 *
Vegetables ^d^ (cup equivalents)	1.4 ± 0.1	2.6 ± 0.2	<0.001 *	86 (45)	65 (34)	0.001 *
Grains ^e^ (ounce equivalents)	7.9 ± 0.3	7.0 ± 0.2	0.17	33 (17)	39 (20)	0.14
Protein ^f^ (ounce equivalents)	3.2 ± 0.2	5.6 ± 0.3	<0.001 *	76 (39)	50 (26)	0.001 *
Dairy ^g^ (cup equivalents)	3.7 ± 0.4	2.5 ± 0.2	0.09	39 (20)	53 (27)	0.26
Oils ^h,i^ (teaspoons)	2.3 ± 0.21	2.4 ± 0.2	0.68	24 (12)	25 (13)	0.82

* Significance at Bonferroni–Holm *p*-value < 0.01. ^a^ Mean intake data presented as mean ± SEM. ^b^ Daily recommendation defined by Dietary Guidelines for Americans for age. Percent not meeting recommendation presented as %(*n*).^c^ The fruit food group includes whole fruits (fresh, canned, frozen, and dried forms) and 100% fruit juice. ^d^ The vegetable food group includes all fresh, frozen, canned, and dried options in cooked or raw forms, including vegetable juices. ^e^ The grains food group includes grains as single foods (e.g., rice, oatmeal, and popcorn), as well as products that include grains as an ingredient (e.g., breads, cereals, crackers, and pasta). Grains are either whole or refined. ^f^ The protein foods group includes foods from both animal and plant sources such as seafood; meats, poultry, and eggs; and nuts, seeds, and soy products. Legumes (beans and peas) are also considered part of the protein foods group as well as the vegetables group. ^g^ Dairy includes all fluid milk and dairy byproducts such as yogurt and cheese, or fortified soy beverages (commonly known as “soymilk”). ^h^ Oils include canola, corn, olive, peanut, safflower, soybean, and sunflower oils. ^i^ Reported as participants exceeding recommendation for fats.

**Table 5 nutrients-11-02013-t005:** Mean energy and macronutrient intake in participants with full or subthreshold ARFID and healthy controls.

	Full or Subthreshold ARFID ^a^	Healthy Controls ^a^	
	*n* = 52	*n* = 52	*p*-Value
Total grams (g)	1970 ± 97	2701 ± 357	0.10
Total Calories (kcal)	2126 ± 80	1967 ± 82	0.24
Total carbohydrates (g)	291 ± 111	252 ± 11.7	0.01 *
Total sugars (g)	131 ± 7.9	107 ± 6.8	0.02
Added sugars (g)	97.0 ± 6.6	66.7 ± 5.4	0.002 *
Total fiber (g)	15.0 ± 0.8	19.1 ± 1.2	0.03
Total protein (g)	63.9 ± 3.2	78.8 ± 3.6	0.003 *
Total fat (g)	82.0 ± 3.6	73.9 ± 3.5	0.17
Solid fats (g)	41.7 ± 2.9	39.1 ± 3.6	0.16

* Significance at Bonferroni-Holm *p*-value < 0.0125. ^a^ Data presented as mean ± SEM.

**Table 6 nutrients-11-02013-t006:** Micronutrient intake including fat-soluble vitamins, water-soluble vitamins, minerals, and trace minerals in participants with full or subthreshold ARFID and healthy controls.

	Mean Intake ^a^			% (*n*) Not Meeting Dietary Reference Intakes ^b^		
	Full or Subthreshold ARFID	Healthy Controls		Full or Subthreshold ARFID	Healthy Controls	
	*n* = 52	*n* = 52	*p*-Value	*n* = 52	*n* = 52	*p*-Value
Vitamin A (mcg) ^c^	699 ± 32.5	807 ± 32.3	0.16	70 (37)	63 (33)	0.09
Vitamin C (mg)	348 ± 88.3	90.4 ± 4.8	0.45	62 (32)	49 (25)	0.08
Vitamin D (mcg) ^d^	5.4 ± 0.2	6.2 ± 0.3	0.11	93 (48)	92 (48)	0.38
Vitamin E (mg)	10.0 ± 0.4	9.8 ± 0.4	0.65	86 (45)	84 (44)	0.95
Vitamin K (mcg)	55.8 ± 1.9	162 ± 12.3	0.01 *	78 (40)	55 (28)	<0.001 *
Vitamin B6 (mg) ^e^	1.6 ± 0.04	1.9 ± 0.04	0.09	38 (20)	27 (14)	0.07
Folate (mcg) ^f^	560 ± 17.2	569 ± 14.5	0.44	41 (21)	32 (17)	0.14
Vitamin B12 (cobalamin, mcg)	3.9 ± 0.1	4.7 ± 0.2	0.01 *	36 (19)	32 (17)	0.12
Calcium (mg)	1096 ± 29.1	1037 ± 26.7	0.78	63 (33)	72 (37)	0.75
Iron (mg)	14.5 ± 0.3	15.7 ± 0.4	0.14	45 (24)	46 (24)	0.38
Magnesium (mg)	248 ± 4.8	299 ± 6.6	0.05	90 (47)	76 (39)	0.002 *
Zinc (mg)	9.4 ± 0.2	11.0 ± 0.3	0.03	65 (34)	52 (27)	0.01 *

* Significance at Bonferroni–Holm *p* -value < 0.0125. ^a^ Mean intake data presented as mean ± SEM. ^b^ Not meeting the recommendation is defined as not meeting age-appropriate recommended dietary allowances (RDA) or Adequate Intake (AI) for males and females, respectively. Data presented as %(*n*). ^c^ Vitamin A defined as Retinol Activity Equivalents (RAE). ^d^ Vitamin D defined as cholecalciferol. ^e^ Vitamin B6 defined as pyridoxine, pyridoxyl, and pyridoxamine. ^f^ Folate as dietary folate equivalents (DFE).

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
