# Peer review of "A Diet High in Processed Foods, Total Carbohydrates and Added Sugars, and Low in Vegetables and Protein Is Characteristic of Youth with Avoidant/Restrictive Food Intake Disorder"

_nutrients, 2019, doi:10.3390/nu11092013_

Round 1
Reviewer 1 Report
The study described presents a food analysis of ARFID. It provides useful information for epidemiologists and clinicians and the findings are well described. A few points to consider: -Who administered the psychiatric interviews? Trained assistants? -How did you define sub-clinical ARFID for the purposes of establishing groups? -the description of the healthy control studies from which subjects were included is good but could be more succinct or put in supplementary methods. -although you describe the demographic results, a table with an indication of differences would be easier to comprehend versus the paragraph. -What is your bonferroni cutoff for each hypothesis? It would be helpful if this was in the methods section. To that, only table 3 lists that the p-value bonfferroniAuthor Response
Please see the attachment.

Reviewer 2 Report
This is a very interesting and comprehensive approach with a well designed experimental procedure. The paper is very well written with great body of evidence. Furthermore, the advanced statistical techniques utilized provide extra valuable information. There are two minor concerns that need to be addressed by the authors.
Do the authors have any data regarding other food constituents than vitamins and proteins? How the results of the present study could be utilized for building therapeutic interventions?
